# Non-Markovian recovery makes complex networks more resilient against large-scale failures

Zhao-Hua Lin[1], Mi Feng [2], Ming Tang [1,2✉], Zonghua Liu[1✉], Chen Xu[3], Pak Ming Hui [4] & Ying-Cheng Lai [5]

Non-Markovian spontaneous recovery processes with a time delay (memory) are ubiquitous in the real world. How does the non-Markovian characteristic affect failure propagation in complex networks? We consider failures due to internal causes at the nodal level and external failures due to an adverse environment, and develop a pair approximation analysis taking into account the two-node correlation. In general, a high failure stationary state can arise, corresponding to large-scale failures that can significantly compromise the functioning of the network. We uncover a striking phenomenon: memory associated with nodal recovery can counter-intuitively make the network more resilient against large-scale failures. In natural systems, the intrinsic non-Markovian characteristic of nodal recovery may thus be one reason for their resilience. In engineering design, incorporating certain non-Markovian features into the network may be beneficial to equipping it with a strong resilient capability to resist catastrophic failures.

[1] State Key Laboratory of Precision Spectroscopy and School of Physics and Electronic Science, East China Normal University, Shanghai 200241, China. [2] Shanghai Key Laboratory of Multidimensional Information Processing, East China Normal University, Shanghai 200241, China. [3] School of Physical Science and Technology, Soochow University, Suzhou 215006, China. [4] Department of Physics, The Chinese University of Hong Kong, Shatin, Hong Kong SAR, China. [5] School of Electrical, Computer and Energy Engineering, Arizona State University, Tempe, AZ 85287, USA. ✉email: tangminghan007@gmail.com; zhliu@phy.ecnu.edu.cn

The dynamics of failure propagation on complex networks constitute an active area of research in network science and engineering with significant and broad applications. This is because the functioning of a modern society relies on the cooperative working of many networked systems such as the electrical power grids, various transportation networks, computer and communication networks, and business networks, but these networks typically possess a complex structure and are vulnerable to failures and intentional attacks. Among the diverse failure scenarios, one of the most severe types is cascading failures[1], where the failure of some nodes would cause their neighbors to fail and the process would propagate to the entire network, disabling a large fraction of the nodes and causing malfunctioning of the system at a large scale[2–15]. Classic examples of cascading failures includes power blackout—the collapse of power grids[5,6], traffic jams[16], and even economic depression[14,17]. Previous studies mostly focused on how cascading failures occur, how network structures and failure propagation are related, and on network robustness and vulnerability to failure propagation[18–22].

A tacit assumption employed in many previous studies of cascading failures is irreversible failure propagation, where a node, if it has failed, cannot recover and is no longer able to function actively. A failed node is then removed from the network completely, including all the links associated with it. There are real-world situations of networked systems, such as financial and transportation networks, where failed nodes can recover from malfunctioning spontaneously after a collapse[23–28]. In general, there are two types of failure-and-recovery scenarios[29]: internal and external. In the first type, a node fails because of internal causes (e.g., the occurrence of some abnormal or undesired dynamical behaviors within the node), which is independent of the states of its neighbors. In this case, the node can recover spontaneously after a period of time. An example is the failure of a company characterized by a drop in its market value due to poor management, followed by recovery due to internal restructuring. The second type is external failures, where a node's failure is externally triggered, e.g., by the failures of its neighboring nodes. After a period of time, as its local "environment" is improved, the node is able to recover spontaneously. The time of recovery depends not only on the specific type of failure-and-recovery mechanism, i.e., whether internal or external, but also on the individual node and its position in the network. For example, for a given node in the network, it may take longer to complete an internal restructuring process to recover from a failure due to an internal than an external cause. Previous computation and mean-field analysis have revealed that cascading dynamics incorporating a failure-and-recovery mechanism can exhibit a rich variety of phenomena such as phase transitions, hysteresis, and phase flipping[29–33]. With respect to the resilience responses of networks, the effects of removing a fraction of nodes and links on network functions were studied[34–37], demonstrating that resilience can be used to characterize the critical functionality of the network with applications in complex infrastructure engineering[36,37].

In spite of the variations in the recovery dynamics across networks or even nodes in the same network, generally the process can be classified into two distinct types: Markovian and non-Markovian. In a Markovian recovery process, an event occurs at a fixed rate and the interevent time follows an exponential distribution[38–40], rendering memoryless the process. On the contrary, a non-Markovian recovery (NMR) process has memory, as the current state of a node depends not only on the most recent state but also on the previous states. In this case, the interevent time distribution is not exponential but typically exhibits a heavy tail. For example, in human activity and interaction dynamics, the occurrences of contacts among the individuals in a social network

can be characteristically non-Markovian, for which there is mounting empirical evidence[41–48]. Non-Markovian type of recovery process also occurs in biochemical reactions[49] and in the financial markets[12,50]. We note that, in the context of spreading dynamics on complex networks, the effects of the non-Markovian process, due to its high relevance to the real world, have attracted growing attention[51–58]. From the point of view of mathematical analysis, incorporating memories into the dynamical process makes analytic treatment challenging.

While the impacts of non-Markovian processes on spreading dynamics have been reasonably well-documented[51–58], there has been little work so far addressing the influence of non-Markovian recovery process on failure propagation dynamics. In this paper, we address this issue systematically through a comparison study of two types of dynamical processes: one with Markovian and another with NMR. In the Markovian recovery (MR) model, failures due to internal and external causes will recover with different constant rates. In the NMR model, such a constant rate cannot be defined. We thus resort to the recovery time. In particular, we assume that the failed nodes due to internal and external causes will take different time to recover, so a memory effect is naturally built into the model. For each model, we develop a mean-field theory and an analysis based on the pair approximation (PA)[29,59–62] that retains the two-node correlation but ignores any correlation of higher orders. Comparing results with numerical simulations indicates that both mean-field theory and PA analysis capture the key features of the failure propagation dynamics qualitatively, but the PA analysis yields results that are in better quantitative agreement with numerics. The counterintuitive and striking phenomenon is then that non-Markovian character with a memory effect makes the network more resilient against large-scale failures. There are two implications. Firstly, in physical, biological, or other natural networked systems, the intrinsic non-Markovian character of nodal recovery may be one reason for resilience of these networks and their existence in a harsh environment. Secondly, in engineering and infrastructure design, incorporating certain non-Markovian features into the network may help strengthen its resilience and robustness.

## Results

**Spontaneous recovery models.** For general failure propagation dynamics on a network, a node can be in one of two states: an active (labeled as $A$-type) state in which the node functions properly and an inactive state ($I$-type) in which the node has failed. To distinguish the causes for a node to become inactive, we label an inactive node due to internal or external failure as $X$-type or $Y$-type, respectively.

In the NMR model, an $A$-type node may fail spontaneously at the rate $\beta_1$ to become an $X$-type node, or it may fail at the rate $\beta_2$ to become a $Y$-type node when the number of its $A$-type neighboring nodes is less than or equal to a threshold integer value $m$ that sets the limit on neighboring support for proper functioning of a node. Without loss of generality, we assume that external failures occur more frequently than internal failures: $\beta_1 < \beta_2$. This is often the case as internal failures can be made less probable by building up the capability of the nodes through better equipment and/or management, while external failures are uncontrollable and more difficult to avoid. For examples, falling stocks may be the result of unanticipated changes in the market rather than poor management. In a road network, failures are caused more often by congestion than by physical failures. Once a node becomes inactive, it takes time $\tau_1$ to recover from an internal failure (when the node is of the $X$-type) or time $\tau_2$ to recover from an external failure (when the node is of the $Y$-type). The non-Markovian characteristic is taken into account through

the incorporation of a memory effect into the model. In particular, the nodes that will recover at time $t$ constitute those that were turned into $X$-type inactive nodes at the time $t - \tau_1$ and those turned into $Y$-type inactive nodes at the time $t - \tau_2$. Here, we assume $\tau_1 > \tau_2$, for the reason that repairing a node or restructuring the management due to the malfunctioning of the node itself would need more time. For example, reorganizing a company or repairing a road often takes more time. The failure processes characterized by the rates $\beta_1$ and $\beta_2$ as well as the recovery processes as determined by $\tau_1$ and $\tau_2$ are schematically illustrated in Fig. 1.

Note that the case of $\tau_1 < \tau_2$ may also arise in the real world. For example, for an infrastructure network in civil engineering, when an earthquake strikes and destroys buildings (nodes), the time to rebuild can be longer than that required for recovering from internal failures, e.g., the collapse of a roof due to some material failure. Our computations of this case yield qualitatively similar results to those in the case of $\tau_1 > \tau_2$—see Supplementary Note 3 for detail.

The MR and NMR models differ only in the recovery processes. In the MR model, an inactive node of the $X$-type or the $Y$-type recovers at a constant rate $\mu_1$ or $\mu_2$, respectively, as illustrated in Fig. 1. Consequently, the number of nodes recovered at time $t$ depends only on the number of inactive nodes of both $X$-type and $Y$-type at the previous time step.

To develop theories for failure propagation on networks with MR or NMR recovery process and to identify the key differences

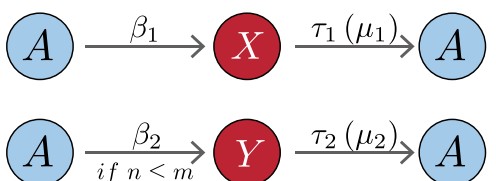

**Fig. 1 Schematic illustration of NMR and MR models.** An active ($A$-type) node may fail spontaneously at the rate $\beta_1$ to become an $X$-type node due to internal causes, or it may fail at the rate $\beta_2$ to become a $Y$-type node when the number of its $A$-type neighbors $n$ is less than or equal to a threshold $m$ setting the necessary neighboring support for the proper functioning of a node. In the NMR model, the $X$-type and $Y$-type nodes take the time duration $\tau_1$ and $\tau_2$ from the time they are generated to recover, respectively. In the MR model, the $X$-type and $Y$-type nodes recover, respectively at the rates $\mu_1$ and $\mu_2$.

between the two type of dynamics, we focus on random regular networks. In the numerical simulations, we use a relatively large network size $N = 3 \times 10^4$ with the degree $k = 35$. In the NMR model, the recovery times are taken to be $\tau_1 = 100$ and $\tau_2 = 1$ for the $X$-type and $Y$-type of nodes, respectively. In the MR model, the values of the recovery rates are set to be $\mu_1 = 1/\tau_1 = 0.01$ and $\mu_2 = 1/\tau_2 = 1$ so that they correspond to the same scales for the recovery times in the NMR model (see Supplementary Note 1 for a more detailed explanation). The threshold values in both models are $m = 15$. Synchronous updating is invoked in simulations with the time step $\Delta t = 0.01$.

**Markovian recovery process.** *Mean-field theory*: We start with setting up the dynamical equations for MR dynamics and comparing results with simulations. Based on the mean-field theory in "Mean-field theory for MR dynamics" of "Methods" section, we first examine the behavior of $E_t([I])$ in Eq. (4). Figure 2a shows the dependence of $E_t([I])$ on the fraction of failed nodes $[I]$. It can be seen that $E_t$ exhibits two different types of behaviors over a large part of $[I]$: $E_t \sim 0$ for a wide range of small $[I]$ values (low failure) and $E_t \sim 1$ for a range of large $[I]$ values (high failure). In the low failure state, external failure events rarely occur. In the high failure state, an active node is supported by an insufficient number of active neighbors and external failure events almost always happen. It implies that the stationary state $[I]$ can possess two branches: setting $E_t = 0$ in Eq. (5) gives $[I] = 1 - 1/(\beta_1/\mu_1 + 1)$ as the low-failure branch, while setting $E_t = 1$ gives $[I] = 1 - 1/(\beta_2/\mu_2 + \beta_1/\mu_1 + 1)$ as the high-failure branch. The two branches are shown in Fig. 2b (dashed and solid curves) in terms of the dependence of $[I]$ on $\beta_1$, for $\mu_1 = 0.01$, $\beta_2 = 2$, and $\mu_2 = 1$ as an example. To check which branch the system would take on and whether there are two states for some range of parameters, the simulation results for moving the value of $\beta_1$ up (circles) and down (squares) are shown in Fig. 2b for comparison. As the values of $\beta_1$ are increased or decreased, the initial state is taken to be the final state corresponding to the previous value of $\beta_1$—the adiabatic process. The results indicate that: (i) the values of $[I]$ from simulations follow the two branches given by the mean-field approximation, and (ii) the low-failure (high-failure) branch is followed when moving $\beta_1$ up (down) until a particular value at which there is a jump to the high-failure (low-failure) branch— the signature of a hysteresis. The results also imply that if one starts from the initial conditions $[X]_0 \neq 0$ and $[Y]_0 = 0$, there exists a critical value of $\beta_c \approx 0.007$ for a sudden increase in the number of failed nodes when $[X]_0$ is small as the system will

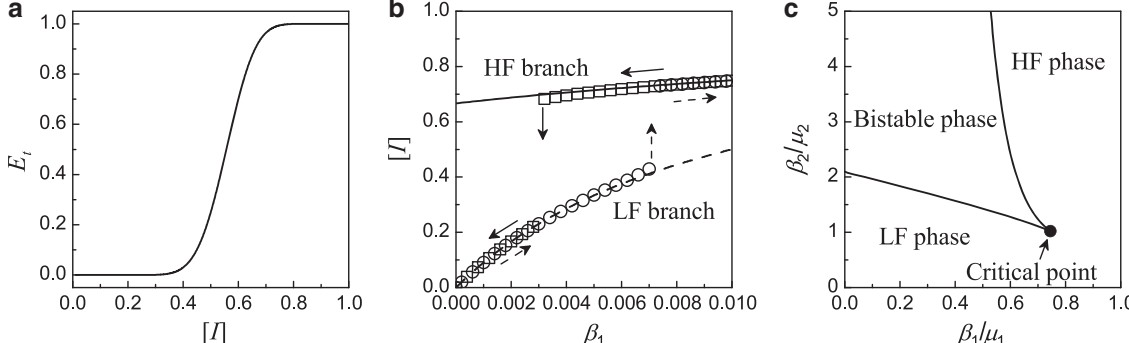

**Fig. 2 Behaviors of MR model. a** Probability $E_t([I])$ in the mean-field theory [Eq. (4)] as a function of $[I]$. **b** Dependence of $[I]$ on $\beta_1$ in the steady state for $\beta_2 = 2.0$, $\mu_1 = 0.01$, and $\mu_2 = 1$. The high-failure (solid curve) and low-failure (dashed curve) branches are calculated by the mean-field theory. The simulation data are obtained by swabbing $\beta_1$ up and down in step of 0.002, starting with $[I]_0 = 0$ for $\beta_1 = 0$. The final state of a value of $\beta_1$ is used as the initial state of the simulations for the next value of $\beta_1$. The arrows indicate the simulation results when the value of $\beta_1$ moves up and down. **c** Phase diagram on the $\beta_2/\mu_2$-$\beta_1/\mu_1$ plane as predicted by the mean-field theory. Systems in the bistable phase will evolve either to a high-failure or a low-failure phase depending on the initial conditions. Beyond the critical point, there is no distinction between the low-failure and high-failure phases.

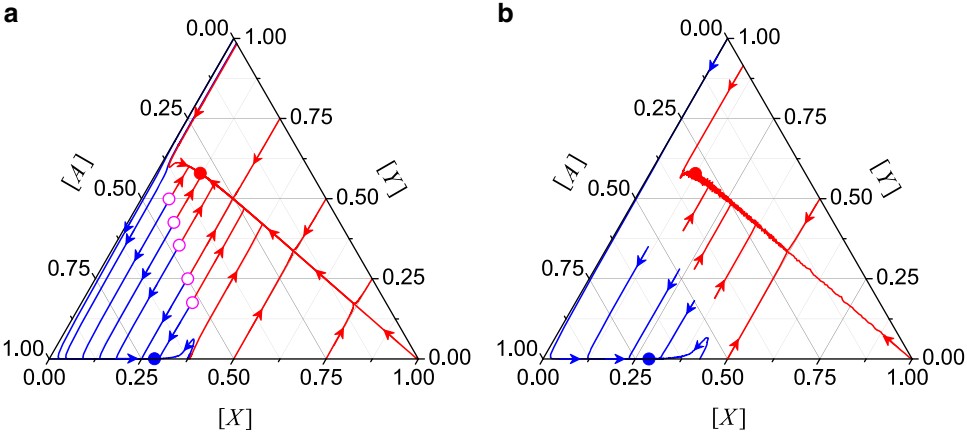

**Fig. 3 Evolutionary properties of MR dynamics.** The dynamical process can be represented as a flow diagram with the lines showing how the fractions of $[X]$, $[Y]$, and $[A]$ evolve in time. The solid circles are the fixed points that the system will finally evolve into. **a** Theoretical calculations based on the mean-field theory. The open circles trace out a separatrix, with systems on different sides evolving into different fixed points. **b** Simulation results. The system parameters are $\beta_1 = 0.004$, $\beta_2 = 2.0$, $\mu_1 = 0.01$, and $\mu_2 = 1$.

follow the low-failure branch. However, for large $[X]_0$, the critical value $\beta_c$ becomes $\beta_c \approx 0.003$ as the system will follow the high-failure branch. A plot of $\beta_c$ against $[X]_0$ will therefore exhibit two plateaus with $\beta_c \approx 0.007$ for small $[X]_0$ and $\beta_c \approx 0.003$ for large $[X]_0$.

The mean-field approximation not only simplifies the analysis but also provides insights into the dynamical process. For example, the mean-field theory suggests the ratios $\beta_1/\mu_1$ and $\beta_2/\mu_2$ as key parameters. In general, solutions can be obtained numerically by solving Eq. (5) together with Eq. (4). The results are shown in Fig. 2c as a phase diagram. For parameters falling into the regions corresponding to the low-failure (high-failure) phase, the system will evolve into a low-failure (high-failure) state. For parameters in the bistable phase, the system will evolve either to a low-failure or a high-failure state, depending on the initial conditions. The high-failure and low-failure phase boundaries meet at the critical point determined by $\beta_1/\mu_1 \approx 0.745$ and $\beta_2/\mu_2 \approx 1.020$.

In addition to the stationary state, the evolution of the system can also be studied by iterating Eqs. (1) and (2) for a given initial condition. Figure 3a shows the evolution of the MR dynamics as obtained by the mean-field theory for $\beta_1 = 0.004$ and $\beta_2 = 2$. In the three-dimensional space formed by $[A]$, $[X]$, and $[Y]$, the sum rule $[A] + [X] + [Y] = 1$ defines a triangular plane, as shown in Fig. 3. At any time $t$, the state of the system is characterized by a point in the plane. The results show that the MR dynamics will evolve into either the low-failure or the high-failure state (filled circles), depending on where the system begins. The mean-field theory also gives a separatrix, the line traced out by the open circles, where the system will evolve into a different state starting from a point on a different side of the separatrix. For $[X]_0 > 0.38$, the system will evolve to a high-failure state with ($[X]$, $[Y]$, $[A]$) given approximately by (0.119, 0.580, 0.301). For $[X]_0 < 0.38$, the system may evolve to the high-failure state or a low-failure state at around (0.285, 0, 0.715). Numerical results are shown in Fig. 3b, verifying all the features predicted by the mean-field theory. For example, the high-failure state is given by ($[X]$, $[Y]$, $[A]$) ~ (0.124, 0.579, 0.298) and the low-failure state at around (0.287, 0, 0.713), both are quite close to the values predicted by the mean-field theory.

*Pairwise approximation theory for the MR model*: It is possible to formulate a theory that takes into account of two-node spatial correlation based on the pairwise approximation (PA). The basic idea is to follow the evolution of different types of links, i.e., links

that connect different pairs of neighboring nodes[62]. The PA method has been used widely in studying epidemic and information spreading[63–65], and in coevolving voter models and adaptive games with two or more strategies[66–69]. In "Effect of nodal correlation: pairwise approximation for the MR model" of "Methods" section, we develop a PA based theory for the MR model.

Figure 4 presents a comparison of the predictions of the PA analysis and mean-field theory with the numerical results, where Fig. 4a shows the time evolution of $[X]_t$ and $[Y]_t$ from the initial state $[X]_0 = [Y]_0 = 0$ for $\beta_1 = 0.009$, $\beta_2 = 2.0$, $\mu_1 = 0.01$, and $\mu_2 = 1$. While both mean-field and PA theories capture the key features in time evolution, the results of PA are in better agreement with those from simulations. It is useful to understand the dynamical behaviors in the MR model qualitatively (so as to enable a meaningful comparison with those of the NMR model later). For this purpose, we identify several stages in the time evolution as marked in Fig. 4a. In the early stage, i.e., $t \in [t_O, t_A]$, most nodes are active and they have more active neighbors, violating the condition $n_A \leq m$. As a result, only internal failures occur and $[X]_t$ grows but $[Y]_t$ decreases and eventually vanishes. For $t \in [t_A, t_B]$, $[X]_t$, active nodes start to fail into $Y$-type nodes, leading to fewer active nodes in the system and triggering more external nodal failures. This results in the observed rapid increase in $[Y]$. In the later stage $t \in [t_B, t_C]$, there are more failed nodes than active ones. While the failed nodes of $X$ and $Y$ types can recover with their respective rates, the remaining or recovered active nodes will more likely fail again through external than internal causes due to the many failed nodes surrounding the active nodes. Consequently, in this later stage, $[Y]_t$ increases and $[X]_t$ decreases toward their respective steady-state values for $t \rightarrow \infty$, with $[Y] > [X]$ when the system evolves into a high-failure state. The PA analysis captures the behavior of $[X]_t$ over time and the onset of $[Y]_t$ better than the mean-field analysis. Figure 4b shows the phase diagram for $\mu_1 = 0.01$ and $\mu_2 = 1.0$. The mean-field phase diagram is the same as that shown in Fig. 2c, where it can be seen that the results of the PA analysis (solid curves) are indeed in better agreement with the simulation results than the predictions of the single-node mean-field theory.

Note that Fig. 2 reveals the emergence of a critical value of $\beta_c$ in the spontaneous failure rate beyond which the system incurs a large-scale failure starting from the initial conditions $[X]_0 \neq 0$ and $[Y]_0 = 0$. The critical rate $\beta_c$ is calculated by starting the system from the initial conditions for different values of $\beta_1$ (for a fixed

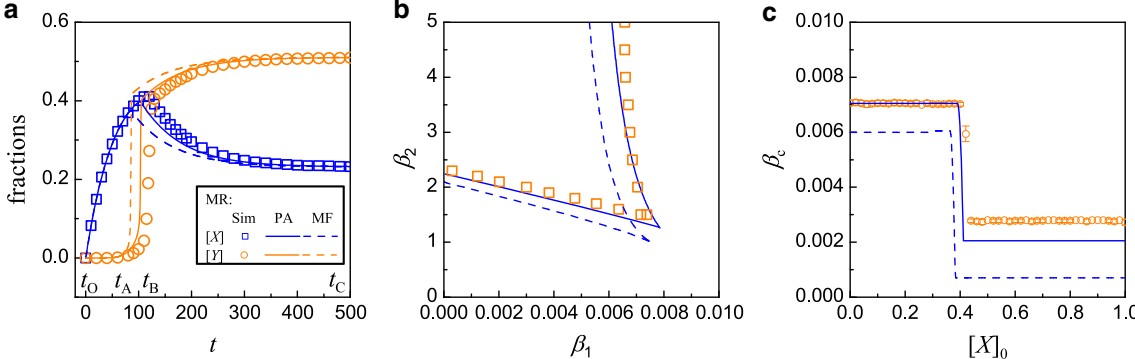

**Fig. 4 Comparison of simulation results with predictions from PA analysis and mean-field theory for the MR model. a** Time evolution of the fraction of inactive nodes. The initial conditions are $[X]_0 = [Y]_0 = 0$. The parameter values are $\beta_1 = 0.009$, $\beta_2 = 2.0$, $\mu_1 = 0.01$, and $\mu_2 = 1.0$. Several time instants are marked to facilitate visualization of the time evolution in different stages: $t_O = 0$, $t_A = 60$, $t_B = 120.7$, and $t_C = 480$. **b** Phase diagram on the $(\beta_2 - \beta_1)$ parameter plane. Other parameters are $\mu_1 = 0.01$, and $\mu_2 = 1.0$. **c** Dependence of $\beta_c$ on the initial value of $[X]_0$, with $[Y]_0 = 0$. The solid (dashed) curve is calculated by the PA (mean-field) theory, and the symbols are for simulation results obtained by averaging over 100 realizations.

value of $\beta_2 = 2.0$) and search for the value of $\beta_1$ beyond which the system attains a high-failure state (see Supplementary Fig. 1 in Supplementary Note 2). The critical value thus depends on $[X]_0$, the initial fraction of failed nodes due to an internal mechanism. Figure 4c shows the numerically obtained functional relation $\beta_c([X]_0)$ (open circles), together with two types of theoretical prediction (PA analysis and mean-field theory). As the initial fraction $[X]_0$ is increased from a near zero value, $\beta_c$ maintains at a relatively higher constant value (about 0.007). As $[X]_0$ increases through the value of about 0.4, the value of the critical rate suddenly decreases to about 0.003. We see that, again, the prediction of the abrupt change in $\beta_c$ by the PA analysis is more accurate than that by the mean-field theory.

What is the physical meaning of the abrupt decrease in the critical value of the spontaneous failure rate as displayed in Fig. 4c? A higher value of $\beta_c$ means that the network system is more resilient to large-scale failures as it requires a larger rate value to drive the system into a high-failure state. As the fraction of initially failed nodes is increased, the network as a whole is more prone to large-scale failure so we expect the value of $\beta_c$ to decrease. Because of the lack of any memory effect in the ideal, Markovian type of recovery process, i.e., after a node fails, it either recovers instantaneously or does not recover (with probabilities determined by the rate of recovery), we expect a characteristic change in the system dynamics as characterized by the value of the critical rate $\beta_c$ to occur in an abrupt manner. Indeed, as Fig. 4c reveals, as the fraction of initially failed nodes is increased through a threshold value, there is a sudden decrease of about 50% in the value of $\beta_c$, giving rise to a first-order type of transition. This behavior of abrupt transition may not occur in reality because of the assumed Markovian recovery process, which is ideal and cannot be expected to arise typically in the physical world. In the next section, we will demonstrate that making the dynamics more physical by assuming non-Markovian type of recovery process will drastically alter the picture of transition in Fig. 4c.

**Non-Markovian recovery process.** To analyze failure propagation dynamics in systems with NMR, a viable approach is to construct difference equations that relate the fractions of types of nodes and links at time $t + \Delta t$ to those at time $t$. It is necessary to keep track of the time when a node becomes the $X$ or $Y$ type as well as the time at which a link becomes type UV. In "Pairwise approximation theory for the NMR model" of "Methods" section, we develop a PA analysis for the NMR model. Figure 5 shows the simulation results from the NMR model, together with the

predictions of the PA analysis and mean-field approximation for $\Delta t = 0.01$. The time evolution of $[X]_t$ and $[Y]_t$ is shown for the parameter setting $\beta_1 = 0.009$, $\beta_2 = 2.0$, $\tau_1 = 100$ (thus $\mu_1 = 0.01$), and $\tau_2 = 1$ (thus $\mu_2 = 1$). The initial conditions are $[X]_0 = [Y]_0 = 0$. Both theories capture the key features of the dynamics. Comparing with results from the MR model [e.g., Fig. 4a], we see that the time evolution of the dynamical variables in the NMR model is different from that in the MR model, in spite of the approximately identical steady-state values.

To describe the key features of the NMR model, we divide the evolution into five stages with the respective time intervals $[t_O, t_A]$, $[t_A, t_B]$, $[t_B, t_C]$, $[t_C, t_D]$, and $[t_D, t_E]$, as shown in Fig. 5a. In the earliest stage $[t_O, t_A]$, $[X]_t$ increases due to internal failures but $[X]_t$ is insufficient to cause external failures. The behavior is similar to that in the MR model, but the duration is shorter and the rise in $[X]_t$ is steeper in the NMR model. The reason is that the memory effect in NMR model allows the recovery of $X$-type nodes to take place only after $\tau_1$ steps, while the recovery occurs probabilistically in the MR model. In the narrow time window of $[t_A, t_B]$, $[X]_t$ attains a level high enough to trigger the onset of many external failures. As a result, the failed nodes constitute the majority in the system and $[A]_t$ decreases sharply, giving rise to the sharp increase in $[Y]_t$. The $Y$-type nodes recover deterministically after $\tau_2$ ($\tau_2 < \tau_1$) into active nodes. In the period $[t_B, t_C]$, the recovery of $Y$-type nodes refuels the system with active nodes that can participate in two paths: more internal and external failures. For $t_C < \tau_1$, the existing $X$-type nodes have yet to recover and $[X]_t$ continues to increase but at a slower pace due to the external failure path, while $[A]_t$ reduces slightly.

In the time window $[t_C, t_D]$, the initial internally failed nodes begin to recover as $t_C > \tau_1$, in addition to the recovery of the $Y$-type nodes. The $A$-type nodes due to recovery will be more likely to become $Y$-type as the failed nodes remain the majority (due to the parameter setting $\beta_2 > \beta_1$ in this example). This leads to the observed increase in $[Y]_t$ and decrease in $[X]_t$ in the time interval $[t_C, t_D]$. In the final stage $[t_D, t_E]$, $[X]_t$ stops decreasing because the recovery of $X$-type nodes at the time $t \gtrsim t_D$ is due to those failed internally at $t \gtrsim t_B$ for which the number was small. However, the recovery of $Y$-type nodes at a shorter time scale supplies fresh active nodes. The fraction of failed nodes $[X]_t + [Y]_t$ is so high, i.e., approaching the high-failure state, that the dynamics lead to a higher steady value of $[Y]$ than $[X]$ in long time. For time well beyond $t_D$, both $[X]$ and $[Y]$ become steady.

Figure 5b shows the phase diagram of the NMR model analogous to Fig. 4b for the MR model, with $\mu_1 = 0.01$ and $\mu_2 = 1.0$. The results of the PA analysis (solid curve) are in better

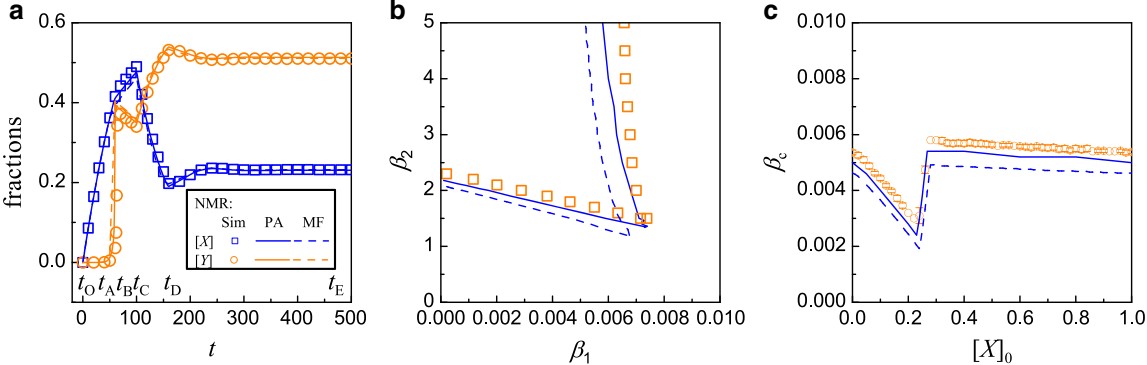

**Fig. 5 Benefit of non-Markovian recovery to making the network more resilient against large-scale failures.** Shown are results from the NMR model through simulations, PA analysis, and mean-field theory. **a** Time evolution of inactive nodes from the initial conditions $[X]_0 = [Y]_0 = 0$, for $\beta_1 = 0.009$, $\beta_2 = 2.0$, $\tau_1 = 100$ (thus $\mu_1 = 0.01$), and $\tau_2 = 1.0$ (thus $\mu_2 = 1.0$). A number of time instants are marked for better visualization of the time evolution in different stages: $t_O = 0$, $t_A = 43.51$, $t_B = 64.68$, $t_C = 100$, $t_D = 164.51$, and $t_E = 480$. **b** Phase diagram in the $(\beta_2 - \beta_1)$ parameter plane for $\tau_1 = 100$ and $\tau_2 = 1.0$. The symbols are numerical results, and the solid and dashed curves are obtained from the PA analysis and mean-field theory, respectively. **c** Dependence of $\beta_c$ for reaching a high-failure state on the initial value of $[X]_0$ with $[Y]_0 = 0$. The error bars with the simulation results are about $6 \times 10^{-5}$, which are obtained by averaging over 100 realizations.

agreement with the simulation results than those obtained from the mean-field theory (dashed curve). The difference in dynamics in the NMR model also alters the dependence of $\beta_c$ to sustain a high-failure state on $[X]_0$. Carrying out the same analysis as for the MR model (see Supplementary Fig. 1 in Supplementary Note 2 for details), we get the relationship $\beta_c([X]_0)$ for attaining a high-failure state for a given initial condition, as shown in Fig. 5c. The pair approximation, again, gives more accurate prediction than that from the mean-field theory.

The result in Fig. 5c demonstrates the striking effect of non-Markovian type of recovery with memory on the failure propagation dynamics, which is in stark contrast to the ideal case of Markovian process as exemplified in Fig. 4c. In particular, as the fraction $[X]_0$ of initially failed nodes is increased from a near zero value to one, the value of $\beta_c$ begins to decrease continuously and smoothly until it reaches a minimum, at which $\beta_c$ increases relatively more rapidly to a high value of about 0.006 for $[X]_0 \approx 0.3$. For $[X]_0 > 0.3$, the value of $\beta_c$ remains approximately constant at 0.006. Comparing Fig. 5c with Fig. 4c, we see two major, characteristic differences. Firstly, the behavior of an abrupt decrease in the Markovian case is replaced by a gradual process in the non-Markovian case, essentially converting a first-order like process to a second-order one. Secondly and more importantly, $\beta_c$ recovers from its minimum value and maintains at a high value regardless of the value of $[X]_0$ insofar as it exceeds about 30%. This means that, the system can maintain its degree of resilience even when the initial fraction of failed nodes reaches 100%! This contrasts squarely the behavior in the Markovian case, where the system resilience is reduced dramatically even when only about 40% of the nodes failed initially. In this sense, we say that a non-Markovian type of memory effect makes the network system more resilient against failure propagation.

While the behavior in Fig. 5c is counterintuitive, a heuristic reason is as follows. For an initial state with many initial $X$-type nodes, the few remaining nodes will switch from being active to the $Y$-type and back. All the initial $X$-type nodes will have to wait for the time period $\tau_1$ to recover. At that time, the system becomes one with only a few failed nodes—effectively equivalent to one with small $[X]_0$ value and requiring a larger $\beta_c$ value to evolve into the high-failure state. In a range of small $[X]_0$, a smaller $\beta_c$ can already cause more active nodes to become $Y$-type, helping maintain the system in a high-failure state as described

for Fig. 5a. Theoretical support for the behavior is provided by the PA analysis and mean-field theory, as shown in Fig. 5c.

In addition to the different time evolution in the MR and NMR models, there are also cases where the same initial conditions $[X]_0$, $[Y]_0$, and $[A]_0$ would lead to different final states. Figure 6 shows the final states starting from any $[X]_0$ and $[Y]_0$ in the $[X]_0 - [Y]_0$ plane (the basin structure), with $\beta_1 = 0.004$, $\beta_2 = 2.0$, $\mu_1 = 0.01$, and $\mu_2 = 1.0$. The results from the mean-field theory (Fig. 6a) and direct simulations (Fig. 6b) show essentially the same features. (Results from the initial-condition setting $[X]_0 \neq 0$ and $[Y]_0 = 0.0$ are presented in Supplementary Fig. 2 of Supplementary Note 2.) It is useful to contrast the final states of the MR and NMR models. From Fig. 3, an initial state, e.g., $[X]_0 = [Y]_0 = 0.5$, will evolve into a high-failure state in the MR model, but it will end up in a low-failure state in the NMR model. This means that, the NMR process can make the system more resilient to failures. (More examples can be found in Supplementary Fig. 3 of Supplementary Note 2 where different steady states from the two models are presented.)

**MR and NMR dynamics on heterogeneous networks.** So far our analysis and simulations have been carried out for MR and NMR dynamics on random regular networks. We find that altering the network structure causes little change in the qualitative results. For example, we have carried out simulations on scale-free networks of size $N = 3 \times 10^4$ with degree range $[k_{min}, \sqrt{N}]$ and degree distribution $P(k) \sim k^{-\gamma}$. Figure 7 shows the results of $\beta_c$ versus $[X]_0$ for the MR and NMR dynamics for networks with $\gamma = 3$. Because of the heterogeneity in the nodal degree distribution, the threshold on external failure is given in terms of the fraction one-half of the failed neighbors.

Comparing results with Fig. 4c for MR dynamics and Fig. 5c for NMR dynamics in random regular networks, we see that the key features are similar when the underlying random regular networks are replaced by scale-free networks. We have also carried out numerical simulations on four additional types of synthetic and empirical networks: (a) networks with degree–degree correlation, (b) networks with a community structure, (c) empirical arenas-email network, and (d) empirical friendship-hamster network, with results presented in Supplementary Notes 4 and 5 for the former and latter two cases, respectively. These results, together with Fig. 7, suggest that, for

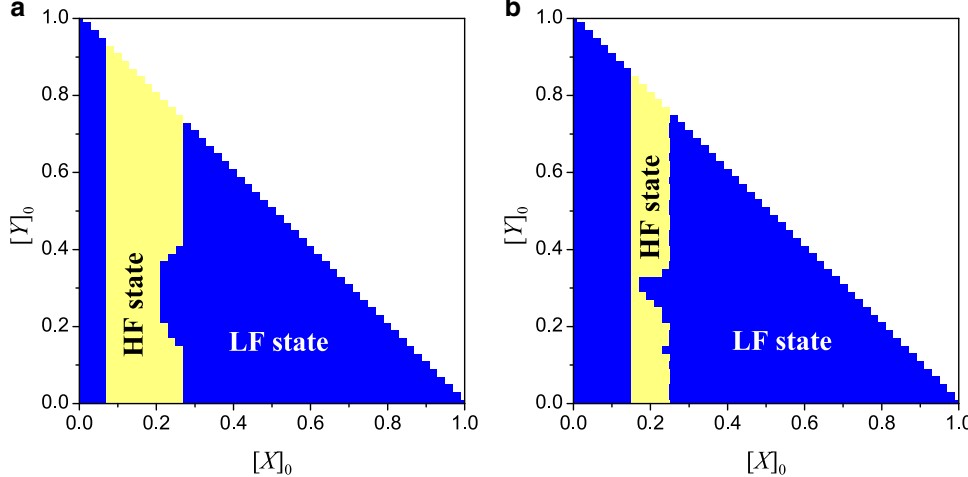

**Fig. 6 Basin structure of NMR model.** On the $[X]_0 - [Y]_0$ plane, basin structure from **a** mean-field theory and **b** simulations. The colors indicate the nature of the steady states given the initial conditions ($[X]_0$, $[Y]_0$). The parameter setting is the same as that in Fig. 3 for the MR model. The simulation results are obtained by averaging over ten statistical realizations.

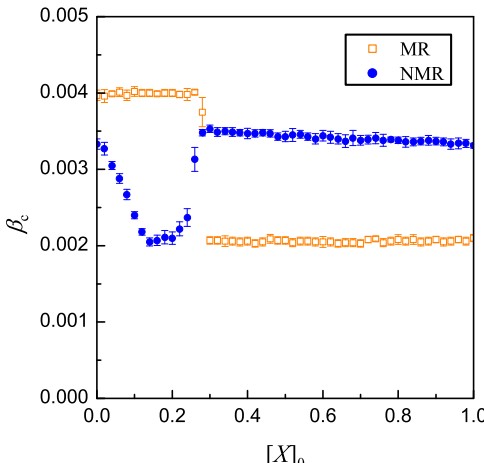

**Fig. 7 MR and NMR dynamics on heterogeneous networks.** The networks are scale-free with $N = 30,000$ nodes, degree exponent $\gamma = 3$, $k_{min} = 6$ and $k_{max} = 173$. Shown is the dependence of $\beta_c$ for reaching a high-failure state on the initial value of $[X]_0$, with $[Y]_0 = 0$, in the MR and NMR models for $\beta_2 = 1.9$, $\mu_1 = 0.01$, $\mu_2 = 1$, $\tau_1 = 100$, and $\tau_2 = 1$. The results qualitatively consistent with those in Fig. 4c for MR dynamics and in Fig. 5c for NMR dynamics on random regular networks.

heterogeneous networks, a non-Markovian process tends to enhance the network resilience against large-scale failures.

## Discussion
The intrinsic memory effect associated with non-Markovian processes makes it challenging to analyze the underlying network dynamics, new and surprising phenomena can arise. Most previous studies treated Markovian processes through either a mean-field type of theory[60,61] or an effective degree approach[59]. For non-Markovian processes, the mean-field approximation can still be applied[29,31–33], but it is necessary to invoke a higher-order theory such as the PA analysis. Our work presents such an example in the context of failure propagation in complex networks.

Our study has demonstrated that, in both models, the network can evolve into a low-failure or a high-failure state, with the latter corresponding to the undesired state of large-scale failure. Both the mean-field and PA theories are capable of predicting the

dynamical behaviors of failure propagation, and the performances of the theories are gauged by simulation results, revealing that the more laborious pair approximation gives results in better quantitative agreement with the numerics. Our systematic computations on different complex networks and two types of theoretical analyses have uncovered a striking phenomenon: the non-Markovian memory effect in the nodal recovery can counterintuitively make the network more resilient against large-scale failures.

Our finding also calls for the incorporation of non-Markovian type of memory factors into the design of communication, computer, and infrastructure networks in various engineering disciplines. We hope our work will stimulate interest in examining and exploiting non-Markovian processes in various network dynamical processes. We have carried out a systematic study of the effects of Markovian versus non-Markovian recovery on network synchronization using the paradigmatic Kuramoto network model, with the main finding that non-Markovian recovery makes the network more resilient against large-scale breakdown of synchronization (Supplementary Note 6).

## Methods
**Mean-field theory for MR dynamics.** Let $[A]_t$, $[X]_t$, and $[Y]_t$ be the fractions of $A$-type, $X$-type, and $Y$-type nodes in the system at time $t$, respectively. A hierarchical set of dynamical equations for the MR model can be constructed to include increasingly longer spatial correlation. The equations for the evolution of the fractions of different types of nodes are:

$$\frac{\mathrm{d}[X]_t}{\mathrm{d}t} = \beta_1 [A]_t - \mu_1 [X]_t, \tag{1}$$

and

$$\frac{\mathrm{d}[Y]_t}{\mathrm{d}t} = \beta_2 E_t [A]_t - \mu_2 [Y]_t, \tag{2}$$

where the first term in each equation gives the supply to $[X]$ ($[Y]$) due to internal (external) failures and the second term represents the drop in $[X]$ ($[Y]$) due to recovery. Note that, because of the relation

$$[A]_t = 1 - [X]_t - [Y]_t \equiv 1 - [I]_t, \tag{3}$$

an equation for $[A]_t$ is unnecessary. The quantity $E_t$ is the probability of an $A$-type node having $j \le m$ neighbors of $A$-type nodes at time $t$ and thus the node will be infected at the rate $\beta_2$.

In general, the quantity $E_t$ involves the correlation between two neighboring nodes. To connect Eqs. (1) and (2) so as to retain the simplicity of a single-node theory, we use the approximation

$$E_t([I]) = \sum_{j=0}^{m} C_k^{k-j} ([I]_t)^{k-j} (1 - [I]_t)^j, \tag{4}$$

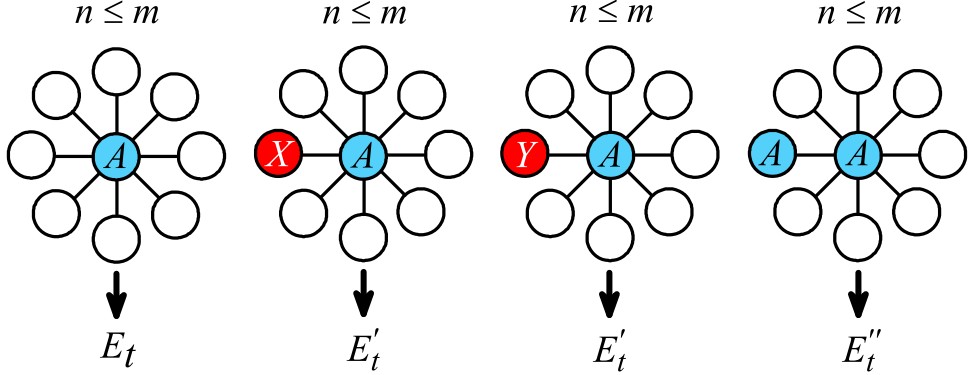

**Fig. 8 Schematic diagram for the PA analysis of the MR model.** The quantities are $E_t$, $E_t'$ and $E_t''$. The blue (red) color indicates a node in the active (failure) state. Open circles are nodes that may take on $A$, $X$, or $Y$ state. In the PA equations, $E_t$ is the probability of an $A$-type node having $n \le m$ neighbors of $A$-type nodes at time $t$, $E_t'$ is the probability of an $A$-type node having $n \le m$ $A$-type neighbors among its $(k - 1)$ neighbors given that one neighbor is inactive, and $E_t''$ is the probability of an $A$-type node having $n \le m - 1$ $A$-type neighbors among its $(k - 1)$ neighbors given that one neighbor is active.

where $C_k^{k-j} = k!/(j!(k - j)!)$. Equations (1)–(4) form a set of equations, from which the fractions of different types of nodes can be solved. This is the simplest single-site mean-field approximation for the MR dynamics that ignores any spatial correlation. Despite its simplicity, it is capable of revealing the key features in the stationary state, in which Eqs. (1) and (2) require the fraction of failed nodes $[I]$ to satisfy

$$[I] = 1 - \frac{1}{(\beta_2/\mu_2)E_t([I]) + (\beta_1/\mu_1) + 1}, \tag{5}$$

which can be solved for $[I]$ self-consistently with Eq. (4). Equation (5) implies that $[I]$ depends only on the ratios $\beta_1/\mu_1$ and $\beta_2/\mu_2$ within the mean-field approximation, and so are the other fractions $[A]$, $[X]$, and $[Y]$.

**Effect of nodal correlation: pairwise approximation for the MR model.** Our PA based analysis begins by defining $[UV]_t$ as the fractions of $UV$ type of links in the system at time $t$, where $U, V \in \{A, X, Y\}$. A connection that stems out from a node can be classified by a type. For example, for a node with the current state being $A$-type, each link that it carries can be classified into the $AA$, $AX$, or $AY$ type, depending on the state of the node at the other end of the link. Taking into account every link from every node, we have that the fractions of links satisfy

$$\sum_{U,V \in \{A,X,Y\}} [UV]_t = 1, \tag{6}$$

with $[UV]_t = [VU]_t$ for $U \ne V$.

In general, the equations of single-node quantities, e.g., Eq. (2), necessarily involve quantities of more extensive spatial correlation because the interplay between the failure of a node and the states of its neighboring nodes. Since $[AI]_t/[A]_t = ([AX]_t + [AY]_t)/[A]_t$ is the probability of an $A$-type node having an inactive node regardless of the types of the neighbors, the probability that there are exactly $j$ neighbors of $A$-type and $(k - j)$ inactive neighbors of either $X$ or $Y$ type is

$$C_k^{k-j} \left( \frac{[AI]_t}{[A]_t} \right)^{k-j} \left( 1 - \frac{[AI]_t}{[A]_t} \right)^j, \tag{7}$$

where $k$ is the degree of the node. The quantity $E_t$ in Eq. 2, as schematically depicted in Fig. 8a, is thus given by

$$E_t = \sum_{j=0}^{m} C_k^{k-j} \left( \frac{[AI]_t}{[A]_t} \right)^{k-j} \left( 1 - \frac{[AI]_t}{[A]_t} \right)^j, \tag{8}$$

which indicates explicitly that the dynamics of single-node quantities are governed by the two-node quantity $[AI]_t$. This is reminiscence of the BBGKY (Bogoliubov-Born-Green-Kirkwood-Yvon) hierarchy of equations for the distribution functions in a system consisting of a large number of interacting particles in statistical physics[70]. Only under the approximation $[AI]_t \approx [A]_t[I]_t$ (so that the two-node correlation can be neglected) will the resulting equation be Eq. (4)—a set of single-node mean-field equations.

To proceed, we derive the dynamical equations for $[UV]_t$ that will in general involve more extensive spatial correlation. For example, a link of the type $AA$ would evolve into a different type depending on the neighborhoods of the two nodes, effectively a small cluster of nodes. To develop a manageable approximation, we retain the two-node correlation and decouple any longer spatial correlation in terms of one-node and two-node functions. This is the idea behind PA for obtaining a closed set of equations. In particular, the dynamical equations for $[AX]_t$

and $[AA]_t$ are

$$\frac{d[AX]_t}{dt} = \mu_1[XX]_t + \mu_2[YX]_t + \beta_1[AA]_t,$$
$$- \mu_1[AX]_t - (\beta_1 + \beta_2 E_t')[AX]_t, \tag{9}$$

and

$$\frac{d[AA]_t}{dt} = 2\mu_1[AX]_t + 2\mu_2[AY]_t,$$
$$- 2(\beta_1 + \beta_2 E_t'')[AA]_t, \tag{10}$$

where

$$E_t' = \sum_{j=0}^{m} C_{k-1}^{k-1-j} \left( \frac{[AI]_t}{[A]_t} \right)^{k-1-j} \left( 1 - \frac{[AI]_t}{[A]_t} \right)^j, \tag{11}$$

is the probability of an $A$-type node having $j \le m$ $A$-type neighbors among its $(k - 1)$ neighbors, given that one neighbor is inactive, and

$$E_t'' = \sum_{j=0}^{m-1} C_{k-1}^{k-1-j} \left( \frac{[AI]_t}{[A]_t} \right)^{k-1-j} \left( 1 - \frac{[AI]_t}{[A]_t} \right)^j, \tag{12}$$

is the probability of an $A$-type node having $j \le m - 1$ $A$-type neighbors among its $(k - 1)$ neighbors, given that one neighbor is active. Figure 8 illustrates the meanings of $E_t$, $E_t'$, and $E_t''$ schematically. The terms in Eqs. (9) and (10) account for how the recovery and failure processes affect the fractions of $AX$-type and $AA$-type links. The complete set of dynamical equations is listed in Supplementary Note 1, which can be solved iteratively to yield the temporal variations on the type of nodes and the type of links given an initial condition. The steady-state quantities can be obtained through a sufficiently large number of iterations.

**Pairwise approximation theory for the NMR model.** Specifically, we let $[U^l]_t$ be the fraction of nodes of type $U$ at time $t$, which became type $U$ from some other type only $l$ time steps ago, and $[U^{l_1} V^{l_2}]_t$ be the fraction of links of the UV type when the corresponding node(s) associated with a link became that of the labeled type $l_1$ and $l_2$ time steps ago. The time evolution of the fraction of $X$-type nodes in the NMR model is given by

$$[X^l]_{t+\Delta t} = \{ \beta_1 \Delta t [A]_t, l \in [0, \Delta t); [X^{l-\Delta t}]_t, l \in [\Delta t, \tau_1]; 0, l \in (\tau_1, \infty).. \tag{13}$$

The first line in Eq. (13) gives the new supply due to internal failure of $A$-type nodes in the time duration $[t, t + \Delta t)$. The second line accounts for the nodes which were inactive for a duration $l - \Delta t$ at time $t$ but have not reached the time for recovery at time $t + \Delta t$. The third line states that all $X$-type nodes that came to existence $\tau_1$ earlier have been recovered. Similarly, the time evolution of the fraction of $Y$-type nodes is given by

$$[Y^l]_{t+\Delta t} = \{ \beta_2 \Delta t E_t [A]_t, l \in [0, \Delta t); [Y^{l-\Delta t}]_t, l \in [\Delta t, \tau_2]; 0, l \in (\tau_2, \infty),. \tag{14}$$

where $E_t$ is defined in Eq. (8) and $[AI]_t = [AX]_t + [AY]_t$. The fractions of $X$-type and $Y$-type nodes, regardless of how long they have been in the corresponding state, are given by $[X]_t = \sum_{l=0}^{\tau_1} [X^l]_t$ and $[Y]_t = \sum_{l=0}^{\tau_2} [Y^l]_t$, respectively. The fraction of active nodes follows from $[A]_t = 1 - [X]_t - [Y]_t$.

To develop a PA analysis for failure propagation dynamics with NMR, we construct the equations for the time evolution of $UV$-types of links and retain

spatial correlation up to two neighboring nodes. Our derivation of the counterparts of Eqs. (13) and (14) in the MR case suggests the necessity to examine the history of the inactive nodes(s) associated with a link. For example, the time evolution of the links in $[AX^l]_t$ is governed by

$$[AX^l]_{t+\Delta t} = \begin{cases} \beta_1 \Delta t [AA]_t + \beta_1 \Delta t ([X^{\tau_1}A]_t + [Y^{\tau_2}A]_t), \\ \qquad\qquad l \in [0, \Delta t); \\ [X^{\tau_1} X^{l-\Delta t}]_t + [Y^{\tau_2} X^{l-\Delta t}]_t \\ \quad + (1 - \beta_1 \Delta t - \beta_2 \Delta t E'_t) \\ \quad \times [AX^{l-\Delta t}]_t, \quad l \in [\Delta t, \tau_1]; \\ 0, l \in (\tau_1, \infty), \end{cases} \quad (15)$$

where $E'$ is defined in Eq. (11). The first line represents the new supply to $AX$-type of links due to an internal failure in one of the active nodes associated with a link of the $AA$-type, and an internal failure together with a recovery of an inactive node in a link of the $XA$-types and $YA$-types. The second line includes the supply to $AX^l$-type links due to recoveries from $XX$ and $YX$ types as well as the links of $AX^{l-\Delta t}$ type that became $AX^l$ type in the recent duration $\Delta t$. The last line comes from the fact that an $X$-type node must recover after a time $\tau_1$ since it became inactive. The fraction of links of $AX$-type, regardless of how long the node in the link has taken in the $X$-type, is given by $[AX]_t = \sum_{l=0}^{\tau_1} [AX^l]_t$. We thus have that the fraction of $AA$-type of links evolves in time as

$$[AA]_{t+\Delta t} = 2(1 - \beta_1 \Delta t - \beta_2 \Delta t E'_t)([AX^{\tau_1}]_t + [AY^{\tau_2}]_t),$$
$$+ [X^{\tau_1} X^{\tau_1}]_t + [Y^{\tau_2} Y^{\tau_2}]_t + 2[X^{\tau_1} Y^{\tau_2}]_t$$
$$+ (1 - 2\beta_1 \Delta t - 2\beta_2 \Delta t E''_t)[AA]_t, \quad (16)$$

where $E''_t$ is defined in Eq. (12). Equations for other types of links can also be constructed (Supplementary Note 1). Equations (15) and (16) are analogous to Eqs. (9) and (10) in the MR model. The number of equations is determined by the divisions of $\tau_1$ and $\tau_2$ into the small time steps $\Delta t$, which increases rapidly when $\Delta t$ is small compared with the other time scales in the NMR dynamics.

A crude approximation analogous to the mean-field theory can be developed for the NMR model by retaining only the fractions of nodes in the equations, which can be done by decoupling the two-node quantities such as $[AI]_t$ by $[AI]_t \approx [A]_t[I]_t$. The resulting equations governing the fractions of different types of nodes become

$$[X]_{t+\Delta t} = \beta_1 \Delta t [A]_t + [X]_t - [X^{\tau_1}]_t, \quad (17)$$

and

$$[Y]_{t+\Delta t} = \beta_2 \Delta t E_t [A]_t + [Y]_t - [Y^{\tau_2}]_t, \quad (18)$$

where $E_t$ takes on the approximate form in Eq. (4). Equations (17), (18), and (4) form a set of equations that can be solved to yield the fractions of different types of nodes. The first two terms in Eqs. (17) and (18) correspond to the increase in inactive nodes due to failure and due to those remaining inactive, and the last term corresponds to recovery. The number of equations, again, depends on the choice of $\Delta t$. This is the mean-field approximation for the NMR model that ignores any spatial correlation.

**Reporting summary**. Further information on research design is available in the Nature Research Reporting Summary linked to this article.

## Data availability

The source data underlying Figs. 2–7 and Supplementary Figs. 1–12 are available at https://github.com/zhlin2328/Codes-for-NCOMMS-19-1125220.

## Code availability

C++ codes to reproduce the data in the main text and the Supplementary Information are available at https://github.com/zhlin2328/Codes-for-NCOMMS-19-1125220.

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

## Acknowledgements

The authors would like to thank Zhenhua Wang for helpful discussions. This work was supported by the National Natural Science Foundation of China (Grant Nos. 11975099, 11575041, 11675056 and 11835003), the Natural Science Foundation of Shanghai (Grant No. 18ZR1412200), and the Science and Technology Commission of Shanghai Municipality (Grant No. 14DZ2260800). Y.C.L. would like to acknowledge support from the Vannevar Bush Faculty Fellowship program sponsored by the Basic Research Office of the Assistant Secretary of Defense for Research and Engineering and funded by the Office of Naval Research through Grant No. N00014-16-1-2828.

## Author contributions

Z.-H.L., M.T., and Z.H.L. designed research; Z.-H.L. performed research; Z.-H.L., M.F., M.T., Z.H.L., C.X., and P.M.H. contributed analytic tools; Z.-H.L., M.F., M.T., Z.H.L., C.X., P.M.H., and Y.-C.L. analyzed data; Z.-H.L., M.T., Z.H.L., P.M.H., and Y.-C.L. wrote the paper.

## Competing interests

The authors declare no competing interests.
