## [Peer Review File · Nature Communications]

Reviewers' Comments:

Reviewer #2:

Remarks to the Author:

This paper is focused on the dynamics of spontaneous recovery on networks in a memoryless (Markovian Recovery or MR) and a memory-dependent (Non-Markovian Recovery or NMR) ways, as well as studies effect of these processes on general network functionality of the networks. The authors address these questions first by setting network structures (regular random and scale-free) with clear rules of transitions for computer simulations, secondly by employing mean-field theory for the evolution of the three classes of nodes and thirdly by employing the pairwise approximation (e.g. Keeling 2005). The latter two, analytical approaches constitute the basis for the development of the MR and NMR exposed.

The results are presented well for the MR case and intuitively expected, exhibiting the various regimes of high and low failure in the network and their interplay with the rate parameters. The counter-intuitive and highlight feature of the work however lies at the NMR process, where, beyond an initial drop in the recovery/resilience of the nodes of the network given an initial fraction of failed nodes, the resilience of the network is maintained even for fractions of failed nodes approaching 100%. The respective regions of high and low failures are again investigated and a comparison and detailed analysis with the mean-field approach is conducted.

The work is certainly novel in tackling the recovery processes of failures in networks in a Markovian and non-Markovian manner and its results have merit. Minor revisions are recommended.

1) In general, the paper lacks citation on defining resilience as network property of systems. Notably, Gao et al. 2016 is a work central in analytical approaches to resilient responses of systems. Similarly for the work of Keeling et al. 1997 is an early reference on pairwise approximation (although the recent ref-erences in pairwise approximation suffice and do make some reference to the line of work of Keeling et al.). Finally, comprehensive discussion of resilience as networked property of complex systems is pro-vided in Ganin, A., et al (2016). Nature Scientific Reports, 6(1), and in Linkov, I., & Trump, B. D. (2019). The Science and Practice of Resilience. Springer, Amsterdam. Conclusion section should pro-vide discussion on strength and limitation of proposed methodology in comparison to other approaches in the field.

2) Regarding the spontaneous recovery processes, $\tau_1 > \tau_2$ is assumed, although external failures may induce equally long recovery, e.g. an earthquake destroying several buildings-nodes would imply more recovery time (i.e. rebuilding time) than some equally serious internal failure (e.g. collapse of a roof of a building due to material failure), which may allow some functionality in the building. This assumption seems to set the stage for the choice of the parameters $\mu_1 - \mu_2$ and $\beta_1 - \beta_2$ later on in the text, both for the MR and NMR processes, and therefore should be justified in greater detail.

3) In equation (10), the authors could expand the right-hand side to one more line so as to make clear that the first two first terms add to 2, i.e. that the pairs $[XA] - [AX]$ and $[YA] - [AY]$ are indeed commutative as previously established (after equation (6) and mentioned in the supplementary note 1). Similarly for the commutative terms in equation (16). In the latter case of (16), I can see in fact that it might make sense not to assume commutative pairs (if the direction of the failure propagation is significant), as recovery takes place in different time intervals ($[\Delta t, \tau_1]$ and $[\Delta t, \tau_2]$). Should this be considered and does it change the results?

4) A few remarks on the justification for the selection of the network parameters when testing the MR and NMR processes on the selected networks (scale-free and regular random) would be enlightening, as the concluding statement "[...] non-Markovian process enhances the network resilience against large-scale failures holds for heterogeneous networks [...]" before VI is rather

bold, not having made tests on empirical networks (I would have presented it as evidence rather than fact). Further, in the legend of figure 8, it is rather meaningless to calculate a mean degree for a scale-free network. I suspect that this went there unintentionally, as it is not mentioned in the main text.

Reviewer #3:

Remarks to the Author:

The authors study the problem of network resilience by considering non-Markovian recovery rates. They verified that non-Markovian node recovery makes the network more resilient against large-scale failures than the Markovian counterpart. Monte Carlo simulations and pair approximation are used in their analysis. The paper is well written and the presented investigation is interesting. However, in my opinion, this paper is an extension of the previous results published by the authors in Nature Communications (10, Article number: 3748 (2019)), where they explored epidemic spreading comparing Markovian and non-Markovian processes. Hence, the results are not so relevant to be published in Nature Communications, but in a more specialized journal. The analysis of random failures and attacks in networks is an old topic and many studies have been performed before. Indeed, this topic is more interesting when dynamical processes are associated, like synchronization of oscillators in power grids. Moreover, they do not consider the more important cases observed in nature, where networks have community organization and degree-degree correlation.

Consequently, I am sorry to say, but the paper should not be accepted. There are many other journals that can be more suitable to them, like Scientific Reports, Proc. Royal. Society, Physical Review E or New Journal of Physics. The paper will find a more interested audience in these journals.

Point-by-point response to referee comments and description of changes made

Reviewer #2

General comments: *“This paper is focused on the dynamics of spontaneous recovery on networks in a memoryless (Markovian Recovery or MR) and a memory-dependent (Non-Markovian Recovery or NMR) ways, as well as studies effect of these processes on general network functionality of the networks. The authors address these questions first by setting network structures (regular random and scale-free) with clear rules of transitions for computer simulations, secondly by employing mean-field theory for the evolution of the three classes of nodes and thirdly by employing the pairwise approximation (e.g. Keeling 2005). The latter two, analytical approaches constitute the basis for the development of the MR and NMR exposed.*

The results are presented well for the MR case and intuitively expected, exhibiting the various regimes of high and low failure in the network and their interplay with the rate parameters. The counter-intuitive and highlight feature of the work however lies at the NMR process, where, beyond an initial drop in the recovery/resilience of the nodes of the network given an initial fraction of failed nodes, the resilience of the network is maintained even for fractions of failed nodes approaching 100%. The respective regions of high and low failures are again investigated and a comparison and detailed analysis with the mean-field approach is conducted.

The work is certainly novel in tackling the recovery processes of failures in networks in a Markovian and non-Markovian manner and its results have merit. Minor revisions are recommended.”

Response: We appreciate that the referee considered our work “certainly novel in tackling the recovery processes of failures in networks in a Markovian and non-Markovian manner and its results have merit.” The referee’s comments have been fully implemented in the revised manuscript.

Minor Comment 1: *“In general, the paper lacks citation on defining resilience as network property of systems. Notably, Gao et al. 2016 is a work central in analytical approaches to resilient responses of systems. Similarly for the work of Keeling et al. 1997 is an early reference on pairwise approximation (although the recent references in pairwise approximation suffice and do make some reference to the line of work of Keeling et al.). Finally, comprehensive discussion of resilience as networked property of complex systems is provided in Ganin, A., et al (2016). Nature Scientific Reports, 6(1), and in Linkov, I., & Trump, B. D. (2019). The Science and Practice of Resilience. Springer, Amsterdam. Conclusion section should provide discussion on strength and limitation of proposed methodology in comparison to other approaches in the field.”*

Response: The references pointed out by the referee are indeed quite relevant to our work, which have been cited as Refs. [41-44] and [96] in the revised manuscript, with the following explanations.

At the end of the second paragraph in Introduction, we have added

- With respect to the resilience response of networks, the effects of removing a fraction of nodes and links on network functions were studied [41-44], demonstrating that resilience

can be used to characterize the critical functionality of the network with applications in complex infrastructure engineering [43,44].

In the first paragraph in Discussion, we have written

- Most previous studies treated Markovian processes through either a mean-field type of theory [94,95] or an effective degree approach [93]. For non-Markovian processes, the mean-field approximation can still be applied [36,38-40] but it is necessary to invoke a higher-order theory such as the PA analysis. Our work presents such an example in the context of failure propagation in complex networks.

Minor comment 2: *“Regarding the spontaneous recovery processes, $\tau_1 > \tau_2$ is assumed, although external failures may induce equally long recovery, e.g. an earthquake destroying several buildings-nodes would imply more recovery time (i.e. rebuilding time) than some equally serious internal failure (e.g. collapse of a roof of a building due to material failure), which may allow some functionality in the building. This assumption seems to set the stage for the choice of the parameters $\mu_1 - \mu_2$ and $\beta_1 - \beta_2$ later on in the text, both for the MR and NMR processes, and therefore should be justified in greater detail.”*

Response: The referee is insightful that it is possible to have $\tau_1 < \tau_2$ in real systems, such as the earthquake example. We have carried out systematic computations to address this issue, with results presented in Supplementary Note 3. In the main text, we have added the following (towards the bottom of page 5):

- Note that, in the real world, the case of $\tau_1 < \tau_2$ can arise. For example, for an infrastructure network in civil engineering, when an earthquake strikes and destroys buildings (nodes), the time to rebuild can be longer than that required for recovering from internal failures, e.g., the collapse of a roof due to some material failure. Our computations of this case yield qualitatively similar results to those in the case of $\tau_1 > \tau_2$ - see Supplementary Note 3 for detail.

In addition, in the main text, we have added the following explanation (towards the bottom of page 4):

- For instance, falling stocks may be the result of unanticipated changes in the market rather than poor management. In a road network, failures are caused more often by congestion than by physical failures. When a rumor spreads through a social network, the impacts on an individual are more likely due to his/her friends than due to spontaneous activities.

A few lines below, we have added

- For example, reorganizing a company or repairing a road often takes more time. Besides, in a social network, individuals spreading the rumor spontaneously tend to stay interested for a longer time.

Minor comment 3: “In equation (10), the authors could expand the right-hand side to one more line so as to make clear that the first two first terms add to 2, i.e. that the pairs $[XA] - [AX]$ and $[YA] - [AY]$ are indeed commutative as previously established (after equation (6) and mentioned in the supplementary note 1). Similarly for the commutative terms in equation (16). In the latter case of (16), I can see in fact that it might make sense not to assume commutative pairs (if the direction of the failure propagation is significant), as recovery takes place in different time intervals ($[\Delta t, \tau_1]$ and $[\Delta t, \tau_2]$). Should this be considered and does it change the results?”

Response: This is a great suggestion. Accordingly, we have modified both Eqs. (10) and (16). Not to assume commutative pairs is indeed justified for homogeneous networks that are used to develop our theory.

Minor comment 4: “A few remarks on the justification for the selection of the network parameters when testing the MR and NMR processes on the selected networks (scale-free and regular random) would be enlightening, as the concluding statement “... non-Markovian process enhances the network resilience against large-scale failures holds for heterogeneous networks ...” before VI is rather bold, not having made tests on empirical networks (I would have presented it as evidence rather than fact). Further, in the legend of figure 8, it is rather meaningless to calculate a mean degree for a scale-free network. I suspect that this went there unintentionally, as it is not mentioned in the main text.”

Response 4: The examples of the scale-free and regular random networks used in the previous version may indeed not be sufficient to support a general conclusion. Following the referee’s comment, we have carried out simulations with four additional examples: (a) networks with degree-degree correlation, (b) networks with a community structure, (c) empirical arenas-email network, and (d) empirical friendship-hamster network, with results presented in Supplementary Notes 4 (a,b) and 5 (c,d). The previous statement in the paragraph preceding the Discussion section has been replaced by the following description:

- We have also carried out numerical simulations on four additional types of synthetic and empirical networks: (a) networks with degree-degree correlation, (b) networks with a community structure, (c) empirical arenas-email network, and (d) empirical friendship-hamster network, with results presented in Supplementary Notes 4 and 5 for the former and latter two cases, respectively. These results, together with Fig. 8, suggest that, for heterogeneous networks, a non-Markovian process tends to enhance the network resilience against large scale failures.

The referee is correct about the mean degree of a scale-free network. The previous statement in the legend of Fig. 8 has been removed and replaced by specification of the minimum and maximum degree.

Reviewer #3

General Comment: *“The authors study the problem of network resilience by considering non-Markovian recovery rates. They verified that non-Markovian node recovery makes the network more resilient against large-scale failures than the Markovian counterpart. Monte Carlo simulations and pair approximation are used in their analysis. The paper is well written and the presented investigation is interesting.”*

Response: We appreciate the referee’s positive statement about our work: “The paper is well written and the presented investigation is interesting.”

Specific Comment 1: *“However, in my opinion, this paper is an extension of the previous results published by the authors in Nature Communications (10, Article number: 3748 (2019)), where they explored epidemic spreading comparing Markovian and non-Markovian processes. Hence, the results are not so relevant to be published in Nature Communications, but in a more specialized journal. The analysis of random failures and attacks in networks is an old topic and many studies have been performed before.”*

Response: Our previous paper entitled “Equivalence and its invalidation between non-Markovian and Markovian spreading dynamics on complex networks” treated a quite different topic than the present manuscript does. There are three major differences between the two papers:

1. The previous paper is on epidemic spreading, while the present one is on cascading failures and recovery. Not only are the intrinsic dynamical processes characteristically different, the associated physical manifestations are also distinct: the former can exhibit second-order transitions while for the latter, the transitions are exclusively first-order.
2. The theoretical approaches used in the two papers are quite different: the previous paper mostly employed a mean-field approach, while the current paper relies on the pair approximation theory so that reinforcement and spontaneous recovery can be treated, with the use of the mean-field approach only for a comparison purpose.
3. Most importantly, the focuses of the two papers are completely independent of each other: the previous paper focuses on the equivalence and invalidation between non-Markovian and Markovian dynamics on epidemic spreading, attempting to unify the two types of dynamics into a general framework. In contrast, the current paper concentrates on how the non-Markovian characteristics influence the resilience performance of a complex networked system, i.e., the effects of non-Markovian spontaneous recovery on cascading failure propagation. To the best of our knowledge, this important topic has not been studied before. A surprising finding here is the counterintuitive phenomenon that the non-Markovian memory process makes the network more resilient against large-scale failures, and this has potential applications in engineering.

While random failures and attacks are an “old topic” with many previous papers, the effects of non-Markovian characteristics on failure propagation dynamics and network resilience have not been studied before. Our present work has revealed that non-Markovian processes have many

new features that do not exist in the Markovian type of cascading failure dynamics. Theoretically, to treat the recovery process of non-Markovian type is extremely challenging, with which our present work has met.

Specific Comment 2: *“Indeed, this topic is more interesting when dynamical processes are associated, like synchronization of oscillators in power grids.”*

Response: This is an excellent suggestion. Accordingly, we have carried out an analysis and computation for a power grid model based on the Kuramoto oscillator and compared the effects of Markovian and non-Markovian recovery on network synchronization. The results are presented in Supplementary Note 6. The general finding is that, in terms of synchronization, non-Markovian recovery makes the network more resilient against large scale breakdown in synchronization. In the main text (in the last paragraph in Discussion), we have added the following statement:

- We have carried out a systematic study of the effects of Markovian versus non-Markovian recovery on network synchronization using the paradigmatic Kuramoto network model, with the main finding that non-Markovian recovery makes the network more resilient against large scale breakdown of synchronization (Supplementary Note 6).

Specific Comment 3: *“Moreover, they do not consider the more important cases observed in nature, where networks have community organization and degree-degree correlation.”*

Response: We have performed systematic computations with degree-degree correlated networks and networks with a community structure. The results have been presented in Supplementary Note 4. In the main text, the following explanation has been given (in the paragraph preceding Discussion):

- We have also carried out numerical simulations on four additional types of synthetic and empirical networks: (a) networks with degree-degree correlation, (b) networks with a community structure, (c) empirical arenas-email network, and (d) empirical friendship-hamster network, with results presented in Supplementary Notes 4 and 5 for the former and latter two cases, respectively. These results, together with Fig. 8, suggest that, for heterogeneous networks, a non-Markovian process tends to enhance the network resilience against large scale failures.

Reviewers' Comments:

Reviewer #2:

Remarks to the Author:

my comments are addressed satisfactory

Reviewer #3:

Remarks to the Author:

The authors improved the paper and answered all of my questions. Thus, now I suggest publication.

Response to Referees

March 25, 2020

Reviewer #2

General comments: *“my comments are addressed satisfactory”*

Response: We truly thank the referee for recommending our paper to publication.

Reviewer #3

General Comment: *“The authors improved the paper and answered all of my questions. Thus, now I suggest publication.”*

Response: We appreciate the referee’s comments and thank the referee for recommending publication.